# Biochar Obtained from *Caryocar brasiliense* Endocarp for Removal of Dyes from the Aqueous Medium

**DOI:** 10.3390/ma15249076

**Published:** 2022-12-19

**Authors:** André L. F. C. Melo, Marcelo T. Carneiro, Ariane M. S. S. Nascimento, Alan I. S. Morais, Roosevelt D. S. Bezerra, Bartolomeu C. Viana, Josy A. Osajima, Edson C. Silva-Filho

**Affiliations:** 1Federal Institute of Piauí, Floriano Campus, Floriano 64808-475, PI, Brazil; 2Interdisciplinary Laboratory for Advanced Materials, Teresina 64049-550, PI, Brazil; 3Federal Institute of Piauí, Teresina-Central Campus, Teresina 64000-040, PI, Brazil

**Keywords:** *Caryocar brasiliense*, adsorption, activated carbon, biochar, aquatic environment

## Abstract

Given the increase in environmental pollution, especially of water, the emergence of studies that seek to develop strategies to mitigate/treat such effects have gained prominence in the world scientific community. Among the numerous adsorption processes, those made from biochar production stand out. This study analyzed the adsorption properties of the blue methylene model dye in the aqueous solution of biochar and activated biochar developed from pequi (*Caryocar brasiliense*) endocarp. The biochar was characterized, before and after adsorption, by infrared spectroscopy (FTIR), scanning electron microscopy (SEM), X-ray diffractometry (XRD), and thermogravimetric analysis (TG). The surface load of the materials was performed by the point of zero charge (pH_PZC_) method. The study also included analyses of contact time parameters and adsorbed concentration in the adsorption process. Morphological analysis showed that a more significant and profound number of fissures and pores appeared in the activated biochar compared to the biochar. Residual mass analysis evidenced that biochar lost about 15% more mass than the activated biochar, indicating that activation occurred satisfactorily. The adsorption process was well adjusted by pseudo-second-order kinetics and Langmuir’s isothermal model. The activated biochar achieved an excellent adsorption capacity of 476.19 mg.g^−1^, thus demonstrating to be a sound system for removing dyes from an aqueous medium.

## 1. Introduction

The current context, marked by the advance of economic activity and worldwide population growth, has led to increased environmental pollution, especially water pollution. It has generated concern and received attention from society and the scientific community [1]. In particular, the growth of industrial activities has contaminated streams, rivers, and seas. Therefore, the academic community and society need to search for innovations that help combat such socioeconomic problems [2,3,4].

A significant contaminant of effluents is wastewater from the dye and paint industries, which should be classified as possible environmental pollutants. The dyes widely used in the textile industry are primarily synthetic and challenging to study or recover. They can generate a series of mutagenic and carcinogenic actions in humans and living organisms in aquatic environments. Pollution caused by dyes, when they react with complexes of chemicals present in the environment, becomes even more resistant to degradation. In particular, studies have shown that Methylene Blue, the dye used in this study, is widely used in the local textile industry in dyeing fabrics [5]. Thus, removing these organic compounds has become a significant technological challenge for the current scientific community [6].

The indiscriminate environmental pollution of dyes by the textile, pharmaceutical, cosmetic, and food industries has become a concern for the scientific community in recent years, which has led to research on techniques for removing these pollutants, with an emphasis on adsorption. The adsorption process is a versatile and affordable alternative for treating various effluents, including those of the textile industry [7,8,9].

One of the most used materials in dye adsorption processes is activated carbon or biochar. It can effectively remove pollutants, mainly organic, in adsorption processes and by the economic viability of production mechanisms and less polluting and generating fewer intermediaries [10,11,12].

Biochars can be produced from materials of plant origin (biomass), which are lignocellulosic, renewable, abundant, and have some degree of porosity [13]. The biomass residues studied for the production of activated carbon include the pequi endocarp (*Caryocar brasiliense*), whose carbonization of biomass via pyrolysis is a necessary thermal modification process [14,15].

The Cerrado biome of Brazil, one of the biomes richest in plant resources, is still minimally used in the daily life of Brazilians. This biome has diverse fruit-producing plant species that humans and animals can consume. Among the fruits of Cerrado, the pequi (*Caryocar brasiliense*) stands out because it has significant economic value through its fruits both in cooking and in the manufacture of liqueurs and medicinal syrups [16,17].

Despite the numerous applications of pequi, only some studies are available about the socioeconomic importance of using its mesocarp, endocarp, and almond. Because it is a typical fruit of the Cerrado, pequi is abundant in several states of Brazil. Finding productive uses for its residues is essential; therefore, research using these as biochar is relevant to the scientific community [18,19,20]. Works involving biochars used in the adsorption process produced from lychee seed, banana, and Moringa oleifera seed can be cited [21,22,23].

This work aimed to develop biochar and activated biochar from biomass residues (pequi endocarp), characterizing them by technical trust and applying them in removing a model organic dye through adsorption tests, varying the time, pH, and concentration parameters.

## 2. Materials and Methods

### 2.1. Materials and Reagents

For this work, the following reagents were used: methylene blue (C_16_H_18_SN_3_Cl, Dinâmica, Brasilpaís), Sodium Hydroxide (NaOH, Dinâmica, 98%, Brazil), Hydrochloric Acid (HCl, Dinâmica, 38%, Brazil), Sodium Chloride (NaCl, Dinâmica, 99%, Brazil), Distilled Water (Brazil). All reagents were analytical in degree, and no previous purification was required. The pequi endocarp residues were obtained at the free fair of the municipality of Floriano (PI), Avenida Bucar Neto, Centro, Floriano (PI), CEP 64800-000.

### 2.2. Biochar Preparation

The biochar was produced in five steps from the base of the pequi endocarp. Initially, pequi was collected, and any other residue was physically removed. In the second step, each part of the pequi (mesocarp, endocarp, and almond) was separated, and the endocarp was dried for 2 h by sunlight (30 ± 2 °C). Then, dried endocarp was crushed in a mill. Third, drying was performed in an oven for 6 h (80 ± 2 °C). In the fourth step, the dry material was placed in porcelain crucibles and charred in a muffle with a heating ratio of 10 °C min^−1^ until reaching the temperature of 500 °C. The vacuum muffle is composed of a closed system, with temperature control, heating, and cooling speed, therefore not allowing oxygen flow. Then, it was cooled to room temperature with a cooling ramp 10 °C min^−1^. The biochar obtained from this process was called biochar of endocarp (BE). 

In the fifth step, the BE underwent a chemical activation with sodium hydroxide (NaOH), in which the ratio between BE and NaOH was 1:3 (m:m). The activation process involved mixing the biochar with the precursor (NaOH). The mixture was submitted to an activation heat treatment, with a heating rate of 5 °C min^−1^ until the temperature of 800 °C. Soon after, the material was submitted to a cooling process with a rate of 5 °C min^−1^. Finally, the material that underwent the activation heat treatment was placed in an agitator with distilled water for 1 h and then filtered the material on qualitative paper, drying it in an oven for 12 h at 105 °C. The activated material was called activated biochar (ABE). NaOH was chosen as a precursor because of its ease of acquisition, economic advantage, and use in works involving activated biochar [21,24,25,26]. 

### 2.3. Biochar Characterizations

The biochars were characterized by Scanning Electron Microscopy (SEM), Infrared Spectroscopy (FTIR), and Thermogravimetry (TG). The infrared spectra were obtained using an Vertex 70 FTIR spectrophotometer (Bruker, MA, USA), by the insert method, with 60 scans in the range of 600 to 4000 cm^−1^. The TG curves were obtained using a thermal analyzer (Shimadzu, TGA-50) (Quioto, Japan) using a heating rate of 10 °C min^−1^ between 25 and 800 °C in an inert nitrogen atmosphere. X-ray diffraction was performed using a Shimadzu instrument, model Labx-XDR 6000 (Quioto, Japan) [3,27,28,29,30].

### 2.4. Point of Zero Charge (pH_PZC_)

The pH_PZC_ method determines the pH by balancing loads between the adsorbent surface and a solution. In the test, 20.0 mg samples of activated carbons were added to 20.0 mL of NaCl (0.1 mol.L^−1^). The pH was adjusted to 2 to 12, adding HCl solution at 1.0 mol.L^−1^ and/or NaOH at 1.0 mol.L^−1^. The initial pH was read using one pH meter. For 24 h, the mixtures were left under constant agitation at 140 rpm at 25 °C. After the described period, the solutions were centrifuged for 10 min at 5000 rpm, and the final pH of the solution was measured. The difference between the initial and final pH was calculated (Δ*pH*) (Equation (1)) and plotted from the obtained data, and then the intersection where Δ*pH* = 0 was determined, corresponding to the point of zero charge [31].
(1)$$\Delta pH=pH_{0}-pH_{f}$$

### 2.5. Influence of pH

The effect of pH on dye adsorption was studied at pHs 4, 7, and 10 in triplicate, in which pH was adjusted with NaOH and HCl solutions. Biochar samples were in contact with 40.0 mL of a solution containing the methylene blue model dye (MB), with a concentration of 300.0 mg.L^−1^. The system was agitated for 24 h. After contact time, the solutions were centrifuged for 1 min at 14,000 rpm and diluted. The new concentrations were determined by reading the aliquots in a UV-vis spectrophotometer (Agilent, Cary 60, Santa Clara, CA, USA) using a pre-established calibration curve (665 nm). Finally, the adsorbed amount *qe* and (mg.g^−1^) and adsorption efficiency *R* (%) were calculated according to Equations (2) and (3), respectively [6,26].
(2)$$q_{e}=\frac{\left(C_{i}-C_{f}\right)}{m}\cdot V$$
(3)$$R=\frac{\left(C_{i}-C_{f}\right)}{C_{i}}\times 100\%$$

$C_{i\;}$ and $C_{f\;}$ represent the initial and final concentrations (mg. L^−1^) of the dye, respectively; m corresponds to the mass of the adsorbent in grams; and *V* is equivalent to the volume in liters of the dye solution used.

### 2.6. Adsorption Kinetics

The kinetic adsorption assay was performed in triplicate at 25 °C, with pH adjusted according to the best adsorption capacity. Different masses of biochar were added in different vials containing 40.0 mL of the dye solution, with a concentration of 100 mg.L^-1^ for biochar (BE) and 800 mg.L^−1^ for activated biochar (ABE). The solutions were constantly shaken at 140 rpm in 10, 20, 40, 60, 120, 220, 260, and 300 min intervals. Next, the samples were centrifuged, and the resulting solutions were measured in a UV-vis spectrophotometer to calculate the concentrations, according to Equation (2). The experimental data were adjusted to two kinetic models (pseudo-first-order and pseudo-second-order) to verify the best fit. The linearized form of the pseudo-first and pseudo-second-order kinetic models are expressed in Equations (4) and (5), respectively [32,33].
(4)$$ln\left(q_{e}-q_{t}\right)=lnq_{e}-k_{1}.t$$
(5)$$\frac{t}{q_{t}}=\frac{1}{k_{2}.q_{e}{}^{2}}+\frac{t}{q_{e}}$$
where *q_e_* and *q_t_* are the amount of dye adsorbed (mg.g^−1^) at equilibrium and time *t* (min), respectively; *k*_1_ represents the adsorption constant of the pseudo-first-order model (min^−1^), and *k*_2_ is the constant of the pseudo-second-order kinetic model (mg.g^−1^min^−1^).

### 2.7. Adsorption Isotherm

The test was performed in triplicate at a temperature of 25 °C, with pH adjusted according to the best adsorption. First, 70.0 mg of biochar was added to 40.0 mL of the dye in solution, varying the concentrations between 50 and 1000 mg.L^−1^, which agitated the best equilibrium time found in the kinetics experiment. Then, the samples were centrifuged, and measurements of UV-vis spectrophotometer solutions were performed to calculate the equilibrium concentrations resulting from adsorption, according to Equation (2). Finally, we investigated which adsorption isotherm model best fits the experimental data (Langmuir or Freundlich). Equation 6 expresses the linearized form of the Langmuir equation, and Equation (7) presents the linearized form of the Freundlich equation [34,35].
(6)$$\frac{C}{q}=\frac{1}{K_{L}\cdot q_{0}}+\frac{C}{q_{0}}$$
(7)$$lnq=lnK_{F}+\frac{1}{n}lnC$$
where *q* is the amount of species adsorbed by mass of the bioadsorbent (mg.g^−1^), *q*_0_ is the maximum amount of species adsorbed by mass of the bioadsorbent (mg.g^−1^), *C* represents the equilibrium concentration of the adsorbent (mg.L^−1^), *K_L_* is the Langmuir adsorption constant related to the chemical balance between adsorbent and adsorbent (mg.L^−1^), *K_F_* is the Freundlich adsorption constant related to adsorption capacity, *n* is the parameter related to the intensity of the adsorption process.

## 3. Results and Discussion

Pequi endocarp was used to generate the biochars, which have yellowish placement, according to the methodology presented, characterizing them as a fine black powder with low density, characteristic of this type of material. The color change after the thermal process indicates the production of biochar. Visually, it was impossible to verify differences between biochars after activation, so several characterizations were made to verify the properties of the materials obtained before and after adsorption.

### 3.1. Characterization of Biochars

Figure 1 presented the results of XRD, FTIR, thermogravimetric analysis, and pH_PZC_ for the studied biochar samples of the endocarp (BE) and activated biochar (ABE).

The FTIR spectra of BE and ABE are presented in Figure 1A. The band around 1600 cm^−1^ is attributed to C=O vibrations in aromatic structures for biochar samples produced from the BE. After activation, the ABE presented a band around 1000 cm^−1^, usually found in oxidized biochars, and attributed to the elongation of C-O vibrations in acids, ethanols, phenols, ethers, and esters. Another band around 690 cm^−1^ is related to Na-O vibration from changes in the activation process, which indicates structural changes in the biochar [36,37,38].

The thermogravimetric analysis (TG) of the biochars is presented in Figure 1B. The TG curves of BE and ABE are b1 and b2, respectively, while those derived from the thermogravimetric curves (DTG) are b3 and b4, respectively. The thermal profile of both is divided into two stages. The first stage occurs around 45 °C for the BE and 53 °C for the ABE. This mass loss is related to water output and physically adsorbed gases. The second stage occurs around 599 °C for BE, with the output of molecules present in the material’s pores, indicating the need for activation. At a temperature above 700 °C and a maximum of 877 °C for ABE, an event with significant degradation velocity indicates the presence of aromatic compounds in the material after the first pyrolysis. Analysis of the residual mass verified that the BE material lost about 15% more mass than the ABE curve, indicating once again that the activation occurred successfully and the need to activate biochars to remove impurities and molecules present in the materials.

The pH of the medium affects the surface charge of the adsorbent and its degree of ionization and affects the adsorbed species. The pH at the point of zero charges (pH_PZC_) determines an index at which a surface tends to be positively or negatively charged as a function of pH. Figure 1C presents the pH_PZC_ curves of the biochar obtained. The pH_PZC_ value of the BE was 5.7, while the pH_PZC_ value of the ABE was 6.4. For pH values below pH_PZC_ of biochars BE (5.7) and ABE (6.4), the surface charge is positive, indicating that adsorption is favored for anionic species, while values above pH_PZC_ favor adsorption of cationic species, which is the case of the Methylene Blue dye, used in the research. The results confirm that the medium’s pH influences the biochar’s surface. Ions (H^+^ or OH^−^) present in the solution can interact with the active sites of the biochars, thus altering their charge balance [6,39]. The TG curves indicate that the presence of molecules, which were removed with activation, caused a greater acidity on the surface of the BE [40].

The study also presents the results of XRD characterizations before and after adsorption. The results can be found in the Appendix A.

Figure 2 shows the SEM images for the BE and activated carbon sample of the ABE studied with a size of 50 μm (A), 20 μm (B), and 10 μm. These images of the morphologies of biochars in different magnitudes will be used to analyze the shape and surface properties of adsorbents.

The BE images are presented in Figure 2A,B when compared to the ABE. Figure 2D,E have the morphology of three-dimensional particles in the form of nonhomogeneous blocks but with a lower pore presentation, which corroborates the data presented in the TG and the pH_PZC_ values, indicating the filling with organic molecules present. Figure 2C,F of the ABE illustrate larger and deeper cracks and pores than in the BE, again indicating the successful activation and corroborating the techniques described above. The nonhomogeneous structures of biochar (ABE) are irregular and may positively influence adsorption [41,42].

Figure 3 shows the relationship between the amount of dye adsorbed by BE (a1) and ABE (a2) as a function of pH. It illustrates the resulting behavior of the amount adsorbed as a function of pH by studying the pH variation of the MB solution. The BE observed a significant increase in the adsorbed amount of MB at pH 10; thus, we decided to use pH 10 for the following tests of this biochar. On the other hand, the most effective removal of MB by the ABE at pH 7 and 10 were similar, so we chose to use the pH 7 of MB for the following tests of that biochar to optimize the process further. In addition, the adsorption was much higher after activation, proving the success of the activation process of the biochar obtained from the endocarp of pequi. As shown in pH_PZC_ before, the increase in pH causes an increase in the negative charge, increasing electrostatic interactions between MB, the cationic dye, and the adsorbents with a negative charge [43].

The kinetic adsorption studies of BE and ABE are presented in Figure 4A and were adjusted by the pseudo-first-order (PFO) and pseudo-second-order (PSO) models. The linear form of these models, Figure 1, was analyzed using the correlation coefficient (R^2^), whose adjustment parameters are presented in Table 1.

Table 1 shows the PSO model describing the process of adsorption of MB by the biochars in both cases due to the high correlation coefficient (R^2^), 0.98920 for BE and 0.99990 for ABE. The results of *q_e_* calculated by the PSO model were close to the experimental values of *q_e_*, thus indicating that the adsorption of the MB dye involved chemical interactions. The analysis showed that the ABE adsorption had a value of *q_e_* equal to 476.19 mg.g^−1^, much higher than that of the BE, with q equal to 7.78 mg.g^−1^.

The adsorption isotherms of biochars are presented in Figure 4B, indicating a maximum adsorption capacity with 65.5 mg.g^−1^ for BE and 432.3 mg.g^−1^ for ABE, indicating a significant increase in adsorption capacity after activation. The isotherms were adjusted against the Langmuir and Freundlich models. The linear form of Langmuir and Freundlich models is presented in Appendix A and was analyzed using the correlation coefficient (R^2^), whose statistical parameters are presented in Table 2.

Table 2 demonstrates that the process of MB adsorption by biochars is better described by the Langmuir isothermal model in both cases due to the values of the correlation coefficient (R^2^), 0.8784 for the BE and 0.9962 for the ABE. Thus, the analysis suggests that the adsorption process occurs homogeneously on the adsorbent surface in the monolayer and by chemical interactions, corroborating the kinetic data.

The experimental values for each biochar (Figure 4B) follow a concave curve marked by a rapid increase in initial concentrations, indicating a strong affinity of the dye to the materials, following a saturation, characteristic of Langmuir isotherms. Balance adsorption isotherms provide relevant information about this process, with concave isotherms being the most commonly used for microporous biochars. The results also indicate that the R^2^ adjustments were similar for both Langmuir and Freundlich in the case of BE; this suggests the presence of residues that promote a mixture between chemical and physical adsorption processes. As for the ABE biochar, which was better adjusted to the Langmuir model, the result suggests chemical adsorption in a monolayer [44,45].

### 3.2. Characterization of Biochars after Adsorption

After the dye adsorption, the materials were characterized to verify possible changes in characterizations and to prove the fixation of the dye in the biochars. The same acronyms were used to facilitate identification, adding A after the acronyms of the material, BE and ABE for biochar after adsorption, and activated biochar after adsorption, respectively.

The FTIR spectra of the materials after adsorption (Figure 5A) presented significant changes in the results. The band around 1600 cm^−1^ attributed to C=O vibrations in aromatic structures was only found for the BE. For ABE, the bands remained around 1000 cm^−1^, which indicates the elongation of C-O vibrations in acids, ethanols, phenols, ethers, and esters, and around 690 cm^−1^ related to Na-O vibration [46].

The thermal analysis of the BE biochar before and after adsorption is presented in Figure 5B. Figure 5B shows that the thermal profile of the biochar after adsorption remained similar to that presented before adsorption. The difference of about 0.95% refers to the dye mass that was adsorbed, which is a small percentage proportional to the amount adsorbed. It corroborates the low adsorption efficiency of the biochar mentioned above, indicating the need for surface activation. The BE and ABE curves (Figure 5C) verify that it has undergone a significant change in mass loss without changing the temperature range. The increase is proportional to the amount of dye adsorbed, proving by this technique that the dye was fixed on the surface and that the amount adsorbed after activation was much higher than that of biochar without activation, resulting in a difference in the mass loss in the final residue of about 18.18%. Thus, the analysis after adsorption strongly indicates the interaction of the dye with the biochars that reinforce the adsorption data and corroborate other techniques used [47,48].

## 4. Conclusions

Biochar of the pequi endocarp was successfully prepared by pyrolysis of biomass at 500 °C. The biochar underwent an efficient activation process, with an activating agent NaOH at 800 °C, proven by the characterizations with alteration in structure, change of thermal profile by removing some impurities present before activation, and alteration of some FTIR bands, due to the removal of some groups with activation. Both were applied in the adsorption of a model dye, with a better fit to the kinetic model of pseudo-second-order, Langmuir isotherm, and adsorption capacity at pH 10 for BE and 7 and 10 for ABE. The activated material, derived from biomass, renewable, and often discarded in nature without any proven use, is a promising adsorbent for the removal of MB from an aqueous environment with a preferential adsorption capacity of 476.19 mg.g^−1^, which revealed the feasibility of using it as a sustainable source to produce high performance activated carbon. It is noticed that the use of biochar produced from the endocarp of pequi presented itself as an excellent option in the adsorption process. Therefore, future work is suggested to analyze the production of biochar from the other compounds of pequi, almond, and mesocarp.

## Figures and Tables

**Figure 1 materials-15-09076-f001:**
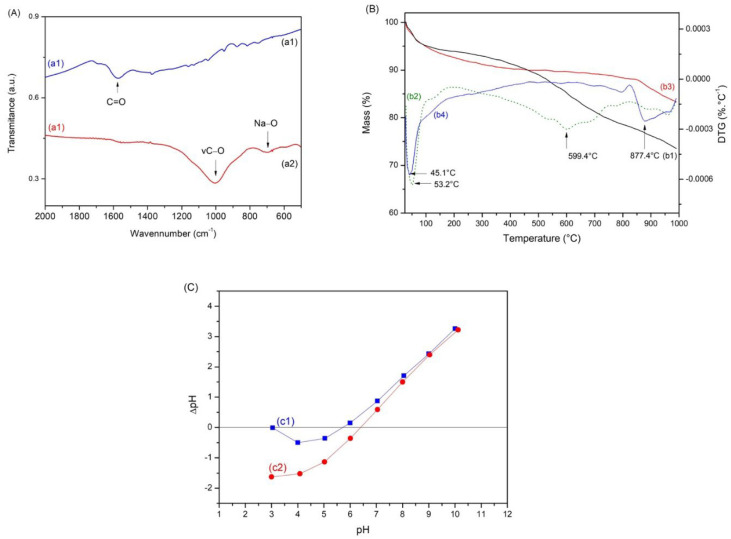
FTIR (**A**) for the studied biochar (BE) (a1) and biochar (ABE) (a2) (**B**); TG and DTG (**B**) curves for the studied samples of biochar (BE), (b1) and (b2), respectively, and curves for the studied samples of biochar (ABE) (b3) and (b4), respectively; (**C**) pH_PZC_ result for the studied samples of biochar (BE) (c1), and biochar (ABE) (c2).

**Figure 2 materials-15-09076-f002:**
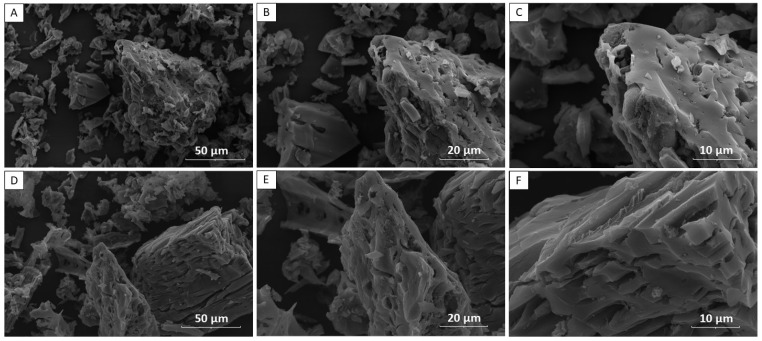
SEM images for the biochar (BE) sample with a size of 50 μm (**A**), 20 μm (**B**), and 20 μm (**C**), and for a sample of biochar (ABE) with a size of 50 μm (**D**), 20 μm (**E**), and 20 μm (**F**).

**Figure 3 materials-15-09076-f003:**
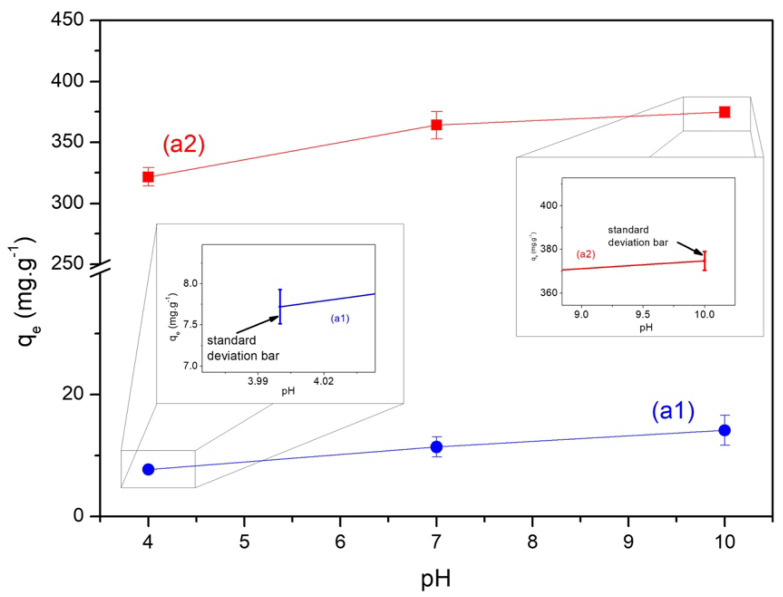
Relationship between the amount of dye adsorbed by the biochar (BE) (a1) and activated biochar (ABE) (a2) depending on the pH.

**Figure 4 materials-15-09076-f004:**
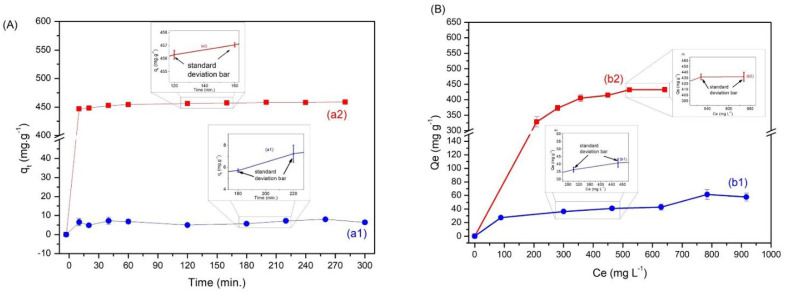
Adsorption kinetics for biochar (BE) (a1) and activated biochar (ABE) (a2) (**A**) and isotherm concentration for biochar (BE) (b1) and activated biochar (ABE) (b2) (**B**).

**Figure 5 materials-15-09076-f005:**
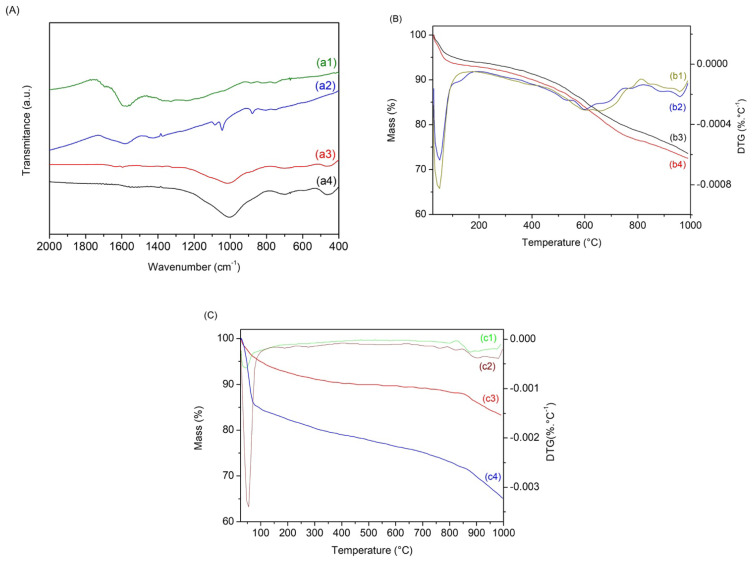
FTIR (**A**) for the studied biochar BE (a2), ABE (a1), BE (a4), and ABE (a3); TG (**B**) curves for the studied samples of biochar BE (b3), ABE (b4) and DTG (**B**) of biochar BE (b2) and ABE (b1) samples; TG (**C**) curves for the studied biochar samples BE (c3), ABE (c4), and DTG (**C**) of biochar samples BE (c1) and ABE (c2).

**Table 1 materials-15-09076-t001:** Adjustment to the kinetic model of pseudo-first-order and pseudo-second-order for the biochar (BE) and activated biochar (ABE).

Biochar	Pseudo-First-Order			Pseudo-Second-Order		
	*q_e_* (mg.g^−1^)	*K*_1_ (min^−1^)	R^2^	*q_e_* (mg.g^−1^)	*K*_2_ (mg.g^−1^.min^−1^)	R^2^
BE	11.20	−0.00982	0.07518	7.78	0.13992	0.9892
ABE	13.46	−0.00700	0.91000	476.19	1.61538	0.9999

**Table 2 materials-15-09076-t002:** Adjustment to Langmuir and Freundlich isotherm model for biochar (BE) and activated biochar (ABE).

Biochar	Langmuir			Freundlich		
	*q_o_*	*K_L_*	R^2^	n	*K_F_*	R^2^
BE	70.92	0.003995	0.8784	3.057	5.93	0.8749
ABE	500.0	0.009809	0.9962	4.091	2.53	0.9105

## Data Availability

Not applicable.

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
