# Peer review of "Biochar Obtained from Caryocar brasiliense Endocarp for Removal of Dyes from the Aqueous Medium"

_materials, 2022, doi:10.3390/ma15249076_

Round 1

Reviewer 1 Report

The paper is about the characterization of Biochar obtained from a Brazilian plant and its application as a dye adsorbent. Generally, the design of the study is scientifically valid, and presentation of the work is good. It needs some improvements prior to consideration for publication.

The title needs to be modified as the authors have not investigated different aquatic environments.

Language needs to be improved.

The application of the dye investigated in this study should be stated in the introduction to have an idea about the level/extent of the problem of this contamination to a broad readership.

Write about the pollution caused by the dyes when they make complexes with the chemicals present in the environment and become more resistant to degradation.

The application of Biochar for dye removal purposes needs to be discussed with respect to the published literature.

What advantages this technique offer over the traditionally used methods for dye adsorption? Compare it with respect to cost, efficiency and feasibility to scale up. Do Biochars favor carbon footprint when their synthesis/formation at high temperature is considered? Generally, in introduction section, the research laps is identified/stated and the significance of the work presented in given in that part. Introduction can be improved accordingly.

While presenting the findings related to characterization of the Biochar, particularly using FTIR, indicate about the possible identification of compounds in the biochar in addition to the functional groups. Related literature can help in this regard.

Discussion is minimal or non-existent. Improve it

State the possible future direction of the research in conclusion section.

Author Response

Journal: Materials

Manuscript:   materials-1961654

Title: Biochar obtained from Caryocar brasiliense endocarp for removal of dyes from aqueous medium

Dear Editor and Reviewers,

Thank you very much for your attention and for the reviewers’ comments on our manuscript ‘Biochar obtained from Caryocar brasiliense endocarp for removal of dyes from aqueous medium (Manuscript materials-1961654)’. We agree with the comments and suggestions, which have been of great assistance in improving the quality of our paper and guiding our research. We have revised the paper after carefully studying the reviewer’s comments and have responded to them point by point. Revisions to the manuscript are highlighted in red. The main revisions and detailed responses to the referees’ comments are listed below. We have also made a number of further changes after carefully rereading the manuscript, which are likewise highlighted in red. 

Referee 1

(1) The paper is about the characterization of Biochar obtained from a Brazilian plant and its application as a dye adsorbent. Generally, the study's design is scientifically valid, and the presentation of the work is good. However, it needs some improvements prior to consideration for publication.

  1. We wanted to thank you for being given us valuable suggestions for the manuscript, which have enriched it. We really appreciate it. Thanks for the encouragements as well as the constructive comments and corrections

(2) The title needs to be modified as the authors have not investigated different aquatic environments.

  1. Thanks for your consideration. The title has been modified as suggestion. Please see the tile.

(3) Language needs to be improved.

  1. Thanks for consideration. Thanks for your consideration. The English language was double-checked as requested.

(4) The application of the dye investigated in this study should be stated in the introduction to have an idea about the level/extent of the problem of this contamination to a broad readership.

  1. Thanks for you observation. The suggestion was made as requested. Please see lines 45, 46 and 47 in the manuscript.

(5) Write about the pollution caused by the dyes when they make complexes with the chemicals present in the environment and become more resistant to degradation.

  1. Thanks for your consideration. The text was modified. Please see lines 45, 46, and 47 in the manuscript.

(6) The application of Biochar for dye removal purposes needs to be discussed with respect to the published literature.

  1. The authors are very grateful for your contribution.The application of Biochar in the adsorption process was discussed in the introduction as requested. Please see lines 74, 75 and 76 of the manuscript.

(7) What advantages this technique offer over the traditionally used methods for dye adsorption? Compare it with respect to cost, efficiency and feasibility to scale up. Do Biochars favor carbon footprint when their synthesis/formation at high temperature is considered? Generally, in introduction section, the research laps is identified/stated and the significance of the work presented in given in that part. Introduction can be improved accordingly.

  1. We appreciate your consideration. The economic feasibility of the process is less polluting and generating less intermediaries. The information was included in the manuscript. Please see lines 57 and 58. The suggested guidelines for modifying the Introduction were made as requested. Please see the Introduction.

(8) While presenting the findings related to characterization of the Biochar, particularly using FTIR, indicate about the possible identification of compounds in the Biochar in addition to the functional groups. Related literature can help in this regard.

  1. Thanks for your consideration. The modifications are done as a suggestion. Please see the manuscript.

(9) Discussion is minimal or non-existent. Improve it

 R: Thanks for your observation.  The discussion has been improved as requested, please see section 3 (Results) in the manuscript.

(10) State the possible future direction of the research in conclusion section.

  1. Thanks for your consideration. The suggestion was made as requested. Please see lines 341 and 342 in the manuscript.

We have tried our best to improve the manuscript. We appreciate the editors’ and reviewers’ work, and hope that the revisions and accompanying responses will make our manuscript suitable for publication in this journal.  Once again, thank you very much for your comments and suggestions.

Yours sincerely, 

Dr  Edson Cavalcanti da Silva Filho

Reviewer 2 Report

In this paper, biochar and activated biochar were developed and characterized from biomass residues (endocarps), and the adsorbability of blue methylene model dye in biochar and active biochar aqueous solution was analyzed, and it was applied to the adsorption test to remove organic dyes and change the time pH Temperature parameters, thus proving that activated biochar is a good system for removing dyes from aqueous media.

Some other comments:

1. The layout of the chart is not neat, and there are cases where the line marks of the chart and the data in the table are inconsistent with the description in the main text (Figure 1, Figure 5, Table 2);

2. The details of the format of references still need attention, and page numbers of some cited references need to be completed;

3. Read through the manuscript and correct some word formatting problems;

4. There is no space between some values and units in the text (e.g. 98% should be changed to 98 %).

Author Response

Journal: Materials

Manuscript:   materials-1961654

Title: Biochar obtained from Caryocar brasiliense endocarp for removal of dyes from aqueous medium

Dear Editor and Reviewers,

Thank you very much for your attention and for the reviewers’ comments on our manuscript ‘Biochar obtained from Caryocar brasiliense endocarp for removal of dyes from aqueous medium (Manuscript materials-1961654)’. We agree with the comments and suggestions, which have been of great assistance in improving the quality of our paper and guiding our research. We have revised the paper after carefully studying the reviewer’s comments and have responded to them point by point. Revisions to the manuscript are highlighted in red. The main revisions and detailed responses to the referees’ comments are listed below. We have also made a number of further changes after carefully rereading the manuscript, which are likewise highlighted in red. 

Referee 2

In this paper, biochar and activated biochar were developed and characterized from biomass residues (endocarps), and the adsorbability of blue methylene model dye in biochar and active biochar aqueous solution was analyzed, and it was applied to the adsorption test to remove organic dyes and change the time pH Temperature parameters, thus proving that activated biochar is a good system for removing dyes from aqueous media.

We wanted to thank you for being given us valuable suggestions for the manuscript, which have enriched it. We really appreciate it. Thanks for the encouragements as well as the constructive comments and corrections.

Some other comments:

(1) The layout of the chart is not neat, and there are cases where the line marks of the chart and the data in the table are inconsistent with the description in the main text (Figure 1, Figure 5, Table 2);

  1. Thank you for your consideration. Inconsistencies between the results of the tables and the writings were corrected, as requested. In addition, the graphics have been improved to Journal. The quality of the original file (.doc) submitted to the Journal is much higher. Please see the manuscript.

(2) The details of the format of references still need attention, and page numbers of some cited references need to be completed;

R: Thank you for your observations. References have been revised and corrected.  Please see the manuscript.

 (3) Read through the manuscript and correct some word formatting problems;

  1. Please accept our apologies. Spelling and formatting corrections were made in the article as requested. Please see the manuscript.

(4) There is no space between some values and units in the text (e.g. 98% should be changed to 98 %).

  1. Thanks for your observation. We double-checked, and the correction was done. Please see the manuscript.

We have tried our best to improve the manuscript. We appreciate the editors’ and reviewers’ work, and hope that the revisions and accompanying responses will make our manuscript suitable for publication in this journal.  Once again, thank you very much for your comments and suggestions.

Yours sincerely, 

Dr  Edson Cavalcanti da Silva Filho

Reviewer 3 Report

View letter

1. The title needs to be simplified.

2. Line 19, coals ? why not use pequi biochar.

3. Keywords lack two  biochar, aquatic environment.

4. Line 32 what about the schematic diagram, belong to the abstract ?

5. Line 76, correct the chemical formula.

6. 2.2 and 2.3 coal meaning ?

7. Table 1 the R2=0.07518 is right ?

8. Line 299, It is difficult to understand that adsorption of organic dyes changes the structure of the BE.

9. Results revised as Results and Discussion.

10. From the results, I could not get the core idea. Can you tell me the mechanism of adsorption, is it chemical or physical or something ? This is very important in the manuscript.

11. Conclusions, the adsorption capacity of 476.19 mg/g is higher or lower, compared with who ?

12. The manuscript need be polished.

13. The TG curves for the studied biochar have any meaning for the analysis ?

Author Response

Journal: Materials

Manuscript:   materials-1961654

Title: Biochar obtained from Caryocar brasiliense endocarp for removal of dyes from aqueous medium

Dear Editor and Reviewers,

Thank you very much for your attention and for the reviewers’ comments on our manuscript ‘Biochar obtained from Caryocar brasiliense endocarp for removal of dyes from aqueous medium (Manuscript materials-1961654)’. We agree with the comments and suggestions, which have been of great assistance in improving the quality of our paper and guiding our research. We have revised the paper after carefully studying the reviewer’s comments and have responded to them point by point. Revisions to the manuscript are highlighted in red. The main revisions and detailed responses to the referees’ comments are listed below. We have also made a number of further changes after carefully rereading the manuscript, which are likewise highlighted in red. 

Referee 3

We wanted to thank you for being given us valuable suggestions for the manuscript, which have enriched it. We really appreciate it. Thanks for the encouragements as well as the constructive comments and corrections.

(1) The title needs to be simplified.

R: R. Thanks for your consideration. The title has been modified as suggestion.  Please see the title.

(2) Line 19, coals ? why not use pequi biochar.

R: Thanks for your observation. The replacement was carried out as requested. Please see the manuscript.

(3) Keywords lack two biochar, aquatic environment.

R: Thanks for observation. Keyword was corrected.  Please see the manuscript.

(4) Line 32 what about the schematic diagram, belong to the abstract ?

  1. Thanks for your consideration. The graphical abstract was removed and will be sent in a separate file according to the Journal guidelines.

(5) Line 76, correct the chemical formula.

  1. Thank you for your considerations. The formula was corrected.

(6) 2.2 and 2.3 coal meaning?

  1. The term “coal” was replaced by “biochar.” Please see the manuscript.

(7) Table 1 the R2=0.07518 is right?

R: Thank you for your consideration. The R2 is correct. The results were calculated and double-checked it. Figure 1A shows the result from the distance of the points with the linearization.

(8) Line 299, It is difficult to understand that adsorption of organic dyes changes the structure of the BE.

  1. The fact that biochar BE has undergone only one heat treatment process leaves weak binding forces in it. In the adsorption process, the weak interactions on the surface of the coal are broken, thus modifying the system's organization. Part of the discussion has been added and can be verified in the manuscript.

(9) Results revised as Results and Discussion.

  1. Thank you for your consideration. We revised the results and Discussion, and the corrections are in the resubmitted manuscript.

(10) From the results, I could not get the core idea. Can you tell me the mechanism of adsorption, is it chemical or physical or something? This is very important in the manuscript.

  1. In the Adsorption Isotherm analysis, the R2 configurations are similar for both Langmuir and Freundlich, suggesting the presence of residues that promote a mixture between the chemical and physical adsorption processes. As for the ABE biochar, which was better adjusted to the Langmuir model, the result suggests chemical adsorption in a monolayer. It was reported in the work as requested by the reviewer.

(11) Conclusions, the adsorption capacity of 476.19 mg/g is higher or lower, compared with who?

  1. The adsorption capacity of 476.19 mg/g was the highest compared to biochars BE and ABE.

(12) The manuscript need be polished.

R: Thank you for your consideration. The manuscript was revised and polished as requested. Please see the manuscript.

(13) The TG curves for the studied biochar have any meaning for the analysis?

  1. The authors are very grateful for your consideration. The TG curves in this work were performed to be one of the characterizations to verify the activation efficiency. To verify that, after the final activation process, gases and other molecules would be removed. After adsorption, the objective was to verify the changes with the incorporation of the dye in the material from the mass variation. The results proved the efficiency of the process, given the difference in apple loss between BE and ABE.

We have tried our best to improve the manuscript. We appreciate the editors’ and reviewers’ work, and hope that the revisions and accompanying responses will make our manuscript suitable for publication in this journal.  Once again, thank you very much for your comments and suggestions.

Yours sincerely, 

Dr  Edson Cavalcanti da Silva Filho

Reviewer 4 Report

 This manuscript brings results of synthesis of new functionalized biochar composite and its applicability as adsorbent for removal of blue methylene. As far as I can determine, the work involved in the study is solid, and the findings are very interesting. Therefore, I believe the paper is appropriate for publication in Materials, after addressing the following comments:

General comments:

 1) pHPCZ was determined in this study for both BE and ABE. However, only the values were given, and no comment was made regarding the relevance of this parameter for the adsorption process. You must add this discussion.

2) You must give an explanation why adsorption efficiency increased with increasing pH. Here is the place where you should use the pHPCZ data.

3) My main concern is around replication. If any replications were performed please highlight them in the methods section and in the graphs.

4) The English of manuscript needs a thorough revision for grammatical errors and typos

Specific comments

 Line 76: ”For the this work”

Delete ”the”

Line 82: ” Coal preparation”

Please replace here and throughout the manuscript, and in the supplementary, the word ”coal” with ”biochar”

Lines 87-89: ” In the fourth step, the dry material was placed in porcelain crucibles and charred in a muffle 88 with a heating ratio 10 °C min-1 until reaching the temperature of 500 °C”

Biochar is produced by pyrolysis, which means heating in absence of oxygen. I suppose you used a nitrogen atmosphere. Please add some details on this matter.

Lines 70-73: ” This work aims to develop biochar and activated biochar from biomass residues pequi endocarp), characterizing them by technical trust, and applying them in the removal of a model organic dye through adsorption tests, varying the time, pH, concentration, and temperature parameters”

The section describing the variation of temperature on adsorption is missing from the submitted manuscript. Please add this section!

Lines 92-93: ” In the fifth step, the BE underwent a chemical activation with sodium hydroxide (NaOH)”

What was the rationale of this activation procedure? Why with NaOH and not with other reagent (e.g., some acid?). Please add all these explanations (and references to support them) in the manuscript.

Lines 93-94: ” The activation process consisted of mixing the biochar with the precursor…”

What precursor? NaOH is the precursor? Please rewrite to become clearer!

Lines 125-127: ” The new concentrations were determined by reading the 125 aliquots in UV-vis spectrophotometer (Agilent, Cary 60) using a pre-established calibra-126 tion curve”

Please give the wavelength (nm) where the absorbance was masured.

Line 267: ”Table 1”

Please add in the manuscript an explanation of the fact that the K2 rate constant of ABE (0.00339) is smaller than of BE (0.01795). From the values of K2 rate constant it results that adsorption with ABE is 5.3 times slower than with BE. Should not be adsorption with ABE faster than with BE?

Author Response

Journal: Materials

Manuscript:   materials-1961654

Title: Biochar obtained from Caryocar brasiliense endocarp for removal of dyes from aqueous medium

Dear Editor and Reviewers,

Thank you very much for your attention and for the reviewers’ comments on our manuscript ‘Biochar obtained from Caryocar brasiliense endocarp for removal of dyes from aqueous medium (Manuscript materials-1961654)’. We agree with the comments and suggestions, which have been of great assistance in improving the quality of our paper and guiding our research. We have revised the paper after carefully studying the reviewer’s comments and have responded to them point by point. Revisions to the manuscript are highlighted in red. The main revisions and detailed responses to the referees’ comments are listed below. We have also made a number of further changes after carefully rereading the manuscript, which are likewise highlighted in red. 

Referee 4

This manuscript brings results of synthesis of new functionalized biochar composite and its applicability as adsorbent for removal of blue methylene. As far as I can determine, the work involved in the study is solid, and the findings are very interesting. Therefore, I believe the paper is appropriate for publication in Materials, after addressing the following comments:

We wanted to thank you for being given us valuable suggestions for the manuscript, which have enriched it. We really appreciate it. Thanks for the encouragements as well as the constructive comments and corrections.

General comments:

(1) pHPCZ was determined in this study for both BE and ABE. However, only the values were given, and no comment was made regarding the relevance of this parameter for the adsorption process. You must add this discussion.

  1. Thank you for your consideration. The following information was added to the work: "The difference between the pH value of BE and ABE points to the success of the activation process. In addition, it indicates greater ease of adsorption of Methylene Blue, a cationic dye, due to the increase in reactions at specific sites due to surface charge".

(2) You must give an explanation why adsorption efficiency increased with increasing pH. Here is the place where you should use the pHPCZ data.

  1. Thanks for your observation. The reviewer's suggestion was accepted and inserted in the resubmitted manuscript.

(3) My main concern is around replication. If any replications were performed please highlight them in the methods section and in the graphs.

  1. Reuse tests were carried out. However, the results still needed improvement, as they probably need a new activation step, as verified by the low adsorption before the activation process—the objective of another work being developed in the research group. For the first test performed, the following methodology was used: The reuse of biochars in the adsorption process was carried out in three stages: adsorption, desorption, and reuse. In adsorption, 0.5 g of carbon was placed in contact with 100.0 mL of MB at a concentration of 600 mg. L-1 under agitation at 130 rpm for 60 min under the conditions determined for the highest adsorption capacity. Then, aliquots were taken and centrifuged at 14000 rpm for 1 min for solid separation. In the desorption, the biomass powders were recovered using the filter paper technique, washed with ethanol, and dried at 40°C for 12 h. After this step, the material is reused. In the question test, 2 (two) reuses were carried out, following the process described above. The adsorption capacity of the materials in the reuse was determined by equation 2, with the dye concentration quantified in the UV-vis Spectrophotometer.

Figure 6: Adsorption of material: (C. E. At.) and (C. E.); Reuse 1: (C. E. At. – R1) and (C. E. – R1); Reuse 2: (C. E. At. – R2) e (C. E. – R2). {C. E. At. is ABE and C. E. is BE}

It is noticed that with each reuse, the materials lose adsorption capacity. Therefore, future studies must remove the dye and activate the material. The results point to a reduction in the efficiency of the adsorption/desorption cycles (under the conditions studied) due to the partial recovery of the adsorptive capacity of the carbons. It is caused by the interaction of specific sorption sites or functional groups found on the surface of coals.

(4) The English of manuscript needs a thorough revision for grammatical errors and typos

  1. Thank you for your consideration. The mansucript was double-cheked.

Specific comments

(5)  Line 76: ”For the this work”

 Delete ”the”

  1. The correction was made. Please see manuscript.

(6) Line 82: ” Coal preparation”

Please replace here and throughout the manuscript, and in the supplementary, the word ”coal” with ”biochar”

  1. The correction was made. Please see manuscript.

(7) Lines 87-89: ” In the fourth step, the dry material was placed in porcelain crucibles and charred in a muffle 88 with a heating ratio 10 °C min-1 until reaching the temperature of 500 °C”

Biochar is produced by pyrolysis, which means heating in absence of oxygen. I suppose you used a nitrogen atmosphere. Please add some details on this matter.

  1. Thank you for your observation. The muffle is composed of a closed system, with temperature control, heating, and cooling speed, without an oxidizing atmosphere.

(8) Lines 70-73: ” This work aims to develop biochar and activated biochar from biomass residues pequi endocarp), characterizing them by technical trust, and applying them in the removal of a model organic dye through adsorption tests, varying the time, pH, concentration, and temperature parameters”

The section describing the variation of temperature on adsorption is missing from the submitted manuscript. Please add this section!

  1. The work needs to present test results with temperature parameters. This information has been corrected in the resubmitted manuscript.

(9) Lines 92-93: ” In the fifth step, the BE underwent a chemical activation with sodium hydroxide (NaOH)”

What was the rationale of this activation procedure? Why with NaOH and not with other reagent (e.g., some acid?). Please add all these explanations (and references to support them) in the manuscript.

  1. NaOH was chosen as a precursor based on the evaluation of some works [21,22]. The compost is accessible, economically advantageous, and used in works involving activated biochar.

(10) Lines 93-94: ” The activation process consisted of mixing the biochar with the precursor…”

What precursor? NaOH is the precursor? Please rewrite to become clearer!

  1. The insertion was performed as requested by the reviewer. Please see the resubmitted manuscript.

(11) Lines 125-127: ” The new concentrations were determined by reading the 125 aliquots in UV-vis spectrophotometer (Agilent, Cary 60) using a pre-established calibra-126 tion curve”

Please give the wavelength (nm) where the absorbance was masured.

  1. The insertion was performed as requested by the reviewer. Please see the resubmitted manuscript.

 (12) Line 267: ”Table 1”

Please add in the manuscript an explanation of the fact that the K2 rate constant of ABE (0.00339) is smaller than of BE (0.01795). From the values of K2 rate constant it results that adsorption with ABE is 5.3 times slower than with BE. Should not be adsorption with ABE faster than with BE?

  1. The calculation of the parameters needed to be corrected. The calculations were redone and are still corrected in the manuscript.

We have tried our best to improve the manuscript. We appreciate the editors’ and reviewers’ work, and hope that the revisions and accompanying responses will make our manuscript suitable for publication in this journal.  Once again, thank you very much for your comments and suggestions.

Yours sincerely, 

Dr  Edson Cavalcanti da Silva Filho

Round 2

Reviewer 2 Report

Accept

Author Response

Journal: Materials

Manuscript:   materials-1961654R2

Title: Biochar obtained in Caryocar brasiliense endocarp for removal of dyes from aqueous medium

Reviewer 2

Accept

Thank you for accepting our paper. We appreciate the editors’ and reviewers’ work.

Yours sincerely, 

Dr  Edson Cavalcanti da Silva Filho

Reviewer 3 Report

1.       The title has two from, so anyone could be replaced with other word (in ?) or the second from revised by in ?  pequi (Caryocar brasiliense) is better ?

2.       Figure 1 caption, the a1, b1, c1, and d1 Integrated into a, because they are one thing. The a2, b2, c2, and d2 Integrated into b, which may make the expression simple and readable. The BE should uniform the same name.

3.       Does the a2 data in Figure 3 have error bar ?

4.       I feel confused about the label as BE and ABE. For example, in Figure 4,  biochar of endocarp (BE) (a1) and activated biochar of the endocarp (ABE) (a2) (A) and isotherm concentration for biochar BE(b1) and ABE (b2) (B). So, the a1 and b1 or a2 and b2 are the same material. Other figures also have the same question, which may mislead or confuse the reader. That is just my personal suggestion.

5.       Line 168, Results and Discussion.

6.       Figure 5, the biochar material is made form crop residue, amorphous not crystal structure and the dyes are organic matter, so the XRD analysis may not important in this article.

Author Response

Journal: Materials

Manuscript:   materials-1961654R2

Title: Biochar obtained in Caryocar brasiliense endocarp for removal of dyes from aqueous medium

Dear Editor and Reviewers,

Thank you very much for your attention and for the reviewers’ comments on our manuscript ‘Biochar obtained in Caryocar brasiliense endocarp for removal of dyes from aqueous medium (Manuscript materials-1961654R2)’. We agree with the comments and suggestions, which have been of great assistance in improving the quality of our paper and guiding our research. We have revised the paper after carefully studying the reviewer’s comments and have responded to them point by point. Revisions to the manuscript are highlighted in red. The main revisions and detailed responses to the referees’ comments are listed below. We have also made a number of further changes after carefully rereading the manuscript, which are likewise highlighted in red. 

Reviewer 3

We wanted to thank you again for being given us valuable suggestions for the manuscript, which have enriched it. We really appreciate it. Thanks for the encouragements as well as the constructive comments and corrections.

  1. The title has two from, so anyone could be replaced with other word (in ?) or the second from revised by in ?  pequi (Caryocar brasiliense) is better?

Response: The correction was made in the manuscript.

  1. Figure 1 caption, the a1, b1, c1, and d1 Integrated into a, because they are one thing. The a2, b2, c2, and d2 Integrated into b, which may make the expression simple and readable. The BE should uniform the same name.

       Response:  As suggested by the reviewer, the table descriptions were reformulated for better understanding by the reader.

  1. Does the a2 data in Figure 3 have error bar?

       Response: Yes. The test was done in triplicate. However, the Journal guidelines requests the graphic as an image, which reduces the quality of the figure. Data tabulation was performed using Origin 8.0 software. The graph with the individual results follows. The graphics were modified to show the error bar in the text.

The graphs were generated from the following results:

Table 1.  Influence of pH 

pH

Biochar (BE)

Biochar (ABE)

Qe(Media)

Er(±)

Qe(Media)

Er(±)

4

7.7210

0.2072

321,5778

7.4507

7

11.4269

1.6535

364,0895

11.1503

10

14.1648

2.4730

374,.7602

4.3162

Table 2. Adsorption Kinetics

Biochar (BE)

Biochar (ABE)

Time (min)

Qe(Media)

Er(±)

Time (min)

Qe(Media)

Er(±)

0

0

0

0

0

0

10

6.52929

1.97923

10

446.5725

0.15796

20

4.84803

0.16747

20

448.23582

1.51678

40

7.18179

1.90093

40

452.53684

0.21505

60

6.84902

1.10707

60

454.25535

0.00381

120

4.99158

0.01522

120

456.29929

0.34256

180

5.70498

0.15877

160

457.05863

0.15796

220

7.23399

0.77429

200

458.2005

0.10467

260

7.96043

0.4176

240

457.88839

0.21125

300

6.40097

1.05052

280

458.91226

0.00952

Table 3. Adsorption Isotherm

Biochar (BE)

Biochar (ABE)

Time (min)

Qe(Media)

Er(±)

Time (min)

Qe(Media)

Er(±)

0

0

0

0

0

0

88,64743

27.26346

1.62899

209.07783

328.56605

15.91426

299,86983

36.41153

1.62737

279.67936

373.10907

8.42947

463,50713

40.84646

2.86907

357.26911

405.19467

10.97285

629,38248

42.84805

4.4141

449.28062

414.35896

0.218

784,13036

61.55045

7.18716

522.00603

432.06929

5.08675

916,73618

57.73153

5.82175

641.98411

432.37873

7.94502

See new Figures 3 and 4 in article.

  1. 4.       I feel confused about the label as BE and ABE. For example, in Figure 4,  biochar of endocarp (BE) (a1) and activated biochar of the endocarp (ABE) (a2) (A) and isotherm concentration for biochar BE(b1) and ABE (b2) (B). So, the a1 and b1 or a2 and b2 are the same material. Other figures also have the same question, which may mislead or confuse the reader. That is just my personal suggestion.

       Response:The reviewer is correct about the difficulty of reading the labels of graphs and tables. The correction and compression of the information on the labels were performed to understand the reader better.

  1. Line 168, Results and Discussion.

Response: The correction was made in the manuscript.

  1. Figure 5, the biochar material is made form crop residue, amorphous not crystal structure and the dyes are organic matter, so the XRD analysis may not important in this article.

       Response: Thanks for the reviewer's suggestion. The characterization test has already been performed. The same has been removed from the Manuscript and added to the supplementary material.

We have tried our best to improve the manuscript. We appreciate the editors’ and reviewers’ work, and hope that the revisions and accompanying responses will make our manuscript suitable for publication in this journal.  Once again, thank you very much for your comments and suggestions.

Yours sincerely, 

Dr  Edson Cavalcanti da Silva Filho

Reviewer 4 Report

The authors present an improved version of their manuscript and they have responded to my earlier comments. However, some of their responses are not satisfactorily. Therefore, I cannot recommend the publication in of this manuscript in Materials. Here are the issues that still need to be addressed by the authors before this manuscript can be published:

General comment 1: The response of authors was:

”The difference between the pH value of BE and ABE points to the success of the activation process. In addition, it indicates greater ease of adsorption of Methylene Blue, a cationic dye, due to the increase in reactions at specific sites due to surface charge”

This response is saying nothing. It has no scientific value.

The point of zero charge indicates the surface charge of adsorbent at different pH values of the solution. You must say here what is the charge of the adsorbent when the pH of the aqueous solution is below the pHzpc?  What is the charge of the adsorbent when the pH of the aqueous solution is greater than the pHzpc? How is the surface charge of adsorbent affecting the adsorption process at pH values below (e.g. 3) and above (e.g. 10) pHzpc, knowing that MB is a cationic dye? It is an advantage or a disadvantage for the adsorption of MB the fact that pHzpc increased from 5.7 (BE) to 6.4 (ABE). All these aspects MUST be discussed in the manuscript when the influence of pH was evaluated.  

General comment 3: My main concern is around replication. If any replications were performed please highlight them in the methods section and in the graphs.

With this comment I am asking you how did you statistically analyzed your experimental data? What software have you used? How many replicates have you performed for each experiment? All the data must represent the mean of at least two independent experiments (two replicates). The standard errors of the means must be indicated by error bars in graphics. You must add these information in the manuscript and the error bars in the graphs!

Specific comments

 Line 82: ” Coal preparation”.

Please replace here and throughout the manuscript, and in the supplementary, the word ”coal” with ”biochar”

The word ”coal” was not replaced with the word ”biochar” throughout the manuscript. This means every occurrence of the word ”coal” in the entire manuscript, including figure captions!

Lines 92-93: ” In the fifth step, the BE underwent a chemical activation with sodium hydroxide (NaOH)”

What was the rationale of this activation procedure? Why with NaOH and not with other reagent (e.g., some acid?). Please add all these explanations (and references to support them) in the manuscript.

Authors provided an explanation and references in the Responses. However, neither the explanation nor the references were added in the text of the manuscript. You must add them in the manuscript!

Author Response

Journal: Materials

Manuscript:   materials-1961654R2

Title: Biochar obtained in Caryocar brasiliense endocarp for removal of dyes from aqueous medium

Dear Editor and Reviewers,

Thank you very much for your attention and for the reviewers’ comments on our manuscript ‘Biochar obtained in Caryocar brasiliense endocarp for removal of dyes from aqueous medium (Manuscript materials-1961654R2)’. We agree with the comments and suggestions, which have been of great assistance in improving the quality of our paper and guiding our research. We have revised the paper after carefully studying the reviewer’s comments and have responded to them point by point. Revisions to the manuscript are highlighted in red. The main revisions and detailed responses to the referees’ comments are listed below. We have also made a number of further changes after carefully rereading the manuscript, which are likewise highlighted in red. 

Reviewer 4

The authors present an improved version of their manuscript and they have responded to my earlier comments. However, some of their responses are not satisfactorily. Therefore, I cannot recommend the publication in of this manuscript in Materials. Here are the issues that still need to be addressed by the authors before this manuscript can be published:

We wanted to thank you again for being given us valuable suggestions for the manuscript, which have enriched it. We really appreciate it. Thanks for the encouragements as well as the constructive comments and corrections.

  • General comment 1: The response of authors was:

”The difference between the pH value of BE and ABE points to the success of the activation process. In addition, it indicates greater ease of adsorption of Methylene Blue, a cationic dye, due to the increase in reactions at specific sites due to surface charge”

This response is saying nothing. It has no scientific value.

The point of zero charge indicates the surface charge of adsorbent at different pH values of the solution. You must say here what is the charge of the adsorbent when the pH of the aqueous solution is below the pHzpc?  What is the charge of the adsorbent when the pH of the aqueous solution is greater than the pHzpc? How is the surface charge of adsorbent affecting the adsorption process at pH values below (e.g. 3) and above (e.g. 10) pHzpc, knowing that MB is a cationic dye? It is an advantage or a disadvantage for the adsorption of MB the fact that pHzpc increased from 5.7 (BE) to 6.4 (ABE). All these aspects MUST be discussed in the manuscript when the influence of pH was evaluated.  

Response: We thank the reviewer for writing guidance in discussing the results regarding the pHPZC. The text was modified in the manuscript. The corrected text follows:

 "The pH of the medium affects the surface charge of the adsorbent and its degree of ionization, and it affects the adsorbed species. The pH at the point of zero charges (pHPZC) determines an index at which a surface tends to be positively or negatively charged as a function of pH. Figure 1D presents the pHPZC curves of the biochar obtained. The pHPZC value of the BE was 5.7, while the pHPZC value of the ABE was 6.4. For pH values ​​below pHPZC of biochars BE (5.7) and ABE (6.4), the surface charge is positive, indicating that adsorption is favored for anionic species, while values ​​above pHpzc favor adsorption of cationic species, which is the case of the Methylene Blue dye, used in the research. The results confirm that the medium's pH influences the biochar's surface. Ions (H+ or OH) present in the solution can interact with the active sites of the biochars, thus altering their charge balance. [6, 40]. The TG curves indicate that the presence of molecules, which were removed with activation, caused a greater acidity on the surface of the BE. [41]”

Regarding the increase in electrostatic interaction, as suggested by the reviewer, it was added in the manuscript:

"As already shown in pHPZC, the increase in pH causes an increase in the negative charge, which increases electrostatic interactions between MB, the cationic dye, and the adsorbents that have a negative charge."

(2) General comment 3: My main concern is around replication. If any replications were performed please highlight them in the methods section and in the graphs.

With this comment I am asking you how did you statistically analyzed your experimental data? What software have you used? How many replicates have you performed for each experiment? All the data must represent the mean of at least two independent experiments (two replicates). The standard errors of the means must be indicated by error bars in graphics. You must add these information in the manuscript and the error bars in the graphs!

Response: The adsorption tests (kinetics and isotherm) and the pH influence test were performed in triplicate. The methodology describes the data; please see lines 126, 139, and 156. The Origin 8.0 software was used to construct the graphics. However, the Journal guideline allows graphics in the form of an image, reducing the figures' quality. Therefore, the graphics were modified to show the text's error bar. The graphs were generated from the following results:

Table 1.  Influence of pH 

pH

Biochar (BE)

Biochar (ABE)

Qe(Media)

Er(±)

Qe(Media)

Er(±)

4

7.7210

0.2072

321,5778

7.4507

7

11.4269

1.6535

364,0895

11.1503

10

14.1648

2.4730

374,.7602

4.3162

Table 2. Adsorption Kinetics

Biochar (BE)

Biochar (ABE)

Time (min)

Qe(Media)

Er(±)

Time (min)

Qe(Media)

Er(±)

0

0

0

0

0

0

10

6.52929

1.97923

10

446.5725

0.15796

20

4.84803

0.16747

20

448.23582

1.51678

40

7.18179

1.90093

40

452.53684

0.21505

60

6.84902

1.10707

60

454.25535

0.00381

120

4.99158

0.01522

120

456.29929

0.34256

180

5.70498

0.15877

160

457.05863

0.15796

220

7.23399

0.77429

200

458.2005

0.10467

260

7.96043

0.4176

240

457.88839

0.21125

300

6.40097

1.05052

280

458.91226

0.00952

Table 3. Adsorption Isotherm

Biochar (BE)

Biochar (ABE)

Time (min)

Qe(Media)

Er(±)

Time (min)

Qe(Media)

Er(±)

0

0

0

0

0

0

88,64743

27.26346

1.62899

209.07783

328.56605

15.91426

299,86983

36.41153

1.62737

279.67936

373.10907

8.42947

463,50713

40.84646

2.86907

357.26911

405.19467

10.97285

629,38248

42.84805

4.4141

449.28062

414.35896

0.218

784,13036

61.55045

7.18716

522.00603

432.06929

5.08675

916,73618

57.73153

5.82175

641.98411

432.37873

7.94502

See new Figures 3 and 4 in article.

Specific comments

(3) Line 82: ” Coal preparation”.

 Please replace here and throughout the manuscript, and in the supplementary, the word ”coal” with ”biochar”

The word ”coal” was not replaced with the word ”biochar” throughout the manuscript. This means every occurrence of the word ”coal” in the entire manuscript, including figure captions!

Response: Per the reviewer's guidance, all the terms "coal" were revised and replaced by "biochar." The changes follow in the manuscript. Please see the manuscript.

(4) Lines 92-93: ” In the fifth step, the BE underwent a chemical activation with sodium hydroxide (NaOH)”

What was the rationale of this activation procedure? Why with NaOH and not with other reagent (e.g., some acid?). Please add all these explanations (and references to support them) in the manuscript.

Authors provided an explanation and references in the Responses. However, neither the explanation nor the references were added in the text of the manuscript. You must add them in the manuscript!

Response: The justification presented was added to the manuscript. Please see the manuscript.

We have tried our best to improve the manuscript. We appreciate the editors’ and reviewers’ work, and hope that the revisions and accompanying responses will make our manuscript suitable for publication in this journal.  Once again, thank you very much for your comments and suggestions.

Yours sincerely, 

Dr  Edson Cavalcanti da Silva Filho

Round 3

Reviewer 3 Report

View letter

1.      Title: Biochar obtained from Caryocar brasiliense endocarp for removal of dyes in the aqueous medium

2.      ABE as activated biochar may be better in the manuscript.

Author Response

Dear Editor and Reviewers,

Thank you very much for your attention and for the reviewers’ comments on our manuscript ‘Biochar obtained from Caryocar brasiliense endocarp for removal of dyes from aqueous medium (Manuscript materials-1961654R3)’. We agree with the comments and suggestions, which have been of great assistance in improving the quality of our paper and guiding our research. We have revised the paper after carefully studying the reviewer’s comments and have responded to them point by point. Revisions to the manuscript are highlighted in red. The main revisions and detailed responses to the referees’ comments are listed below. We have also made a number of further changes after carefully rereading the manuscript, which are likewise highlighted in red. 

Reviewer 3

  1. Title: Biochar obtained from Caryocar brasiliense endocarp for removal of dyes in the aqueous medium

Response: Thanks for suggestions. The title was modified.

  1. ABE as activated biochar may be better in the manuscript.

Response: Thanks for suggestions. The text was standardized.

We have tried our best to improve the manuscript. We appreciate the editors’ and reviewers’ work, and hope that the revisions and accompanying responses will make our manuscript suitable for publication in this journal.  Once again, thank you very much for your comments and suggestions.

Yours sincerely, 

Dr  Edson Cavalcanti da Silva Filho

Reviewer 4 Report

The authors present an improved version of their manuscript and they have responded to all my earlier comments. Therefore, I recommend the publication of this manuscript in Materials. However, I have noticed that a change has been made in the title of the article, from : ”Biochar obtained from Caryocar brasiliense endocarp….” to ”Biochar obtained in Caryocar brasiliense endocarp”. I do not know who made this recommendation, but it is totally wrong. Was the biochar obtained inside the endocarp? I do not believe. Therefore, I recommend using the original title: ”Biochar obtained from Caryocar brasiliense endocarp….”. This is in accord with the final sentence of the Introduction section, which is correctly saying that: ”This work aimed to develop biochar and activated biochar from biomass residues (pequi endocarp)……”

Author Response

The authors present an improved version of their manuscript and they have responded to all my earlier comments. Therefore, I recommend the publication of this manuscript in Materials. However, I have noticed that a change has been made in the title of the article, from :

Response: Thanks for your considerations.

  1. ”Biochar obtained from Caryocar brasiliense endocarp….” to ”Biochar obtained in Caryocar brasiliense endocarp”. I do not know who made this recommendation, but it is totally wrong. Was the biochar obtained inside the endocarp? I do not believe. Therefore, I recommend using the original title: ”Biochar obtained from Caryocar brasiliense endocarp….”. This is in accord with the final sentence of the Introduction section, which is correctly saying that: ”This work aimed to develop biochar and activated biochar from biomass residues (pequi endocarp)……”

Response: Thanks for consideration. The title was adjusted.

We have tried our best to improve the manuscript. We appreciate the editors’ and reviewers’ work, and hope that the revisions and accompanying responses will make our manuscript suitable for publication in this journal.  Once again, thank you very much for your comments and suggestions.

Yours sincerely, 

Dr  Edson Cavalcanti da Silva Filho
